# Organocatalytic Ring-Opening Polymerization of *ε*-Caprolactone Using *bis*(*N*-(*N*′-butylimidazolium)alkane Dicationic Ionic Liquids as the Metal-Free Catalysts: Polymer Synthesis, Kinetics and DFT Mechanistic Study

**DOI:** 10.3390/polym13244290

**Published:** 2021-12-08

**Authors:** Nathaporn Cheechana, Wachara Benchaphanthawee, Natthapol Akkravijitkul, Puracheth Rithchumpon, Thiti Junpirom, Wanich Limwanich, Winita Punyodom, Nawee Kungwan, Chanisorn Ngaojampa, Praput Thavornyutikarn, Puttinan Meepowpan

**Affiliations:** 1Department of Chemistry, Faculty of Science, Chiang Mai University, 239 Huay Kaew Road, Chiang Mai 50200, Thailand; nathaporn_c@cmu.ac.th (N.C.); wachara.bw@gmail.com (W.B.); natthapol_ak@cmu.ac.th (N.A.); puracheth_rith@cmu.ac.th (P.R.); thiti.j@cmu.ac.th (T.J.); winitacmu@gmail.com (W.P.); naweekung@gmail.com (N.K.); chanisorn.ngao@cmu.ac.th (C.N.); tpraput@gmail.com (P.T.); 2Graduate School, Chiang Mai University, 239 Huay Kaew Road, Chiang Mai 50200, Thailand; 3Faculty of Sciences and Agricultural Technology, Rajamangala University of Technology Lanna, 128 Huay Kaew Road, Chiang Mai 50300, Thailand; wanich.lim@gmail.com; 4Center of Excellence for Innovation in Chemistry (PERCH-CIC), Faculty of Science, Chiang Mai University, 239 Huay Kaew Road, Chiang Mai 50200, Thailand; 5Center of Excellence in Materials Science and Technology, Chiang Mai University, 239 Huay Kaew Road, Chiang Mai 50200, Thailand

**Keywords:** *bis*(*n*-(*n*′-butylimidazolium)alkane dicationic ionic liquids, ring-opening polymerization, *ε*-caprolactone, metal-free catalyst, density functional theory

## Abstract

In this work, we successfully synthesized high thermal stable 1,n-*bis*(*N*-(*N*′-butylimidazolium)alkane *bis*hexafluorophosphates (1,n-*bis*[Bim][PF_6_], n = 4, 6, 8, and 10) catalysts in 55–70% yields from imidazole which were applied as non-toxic DILs catalysts with 1-butanol as initiator for the bulk ROP of *ε*-caprolactone (CL) in the varied ratio of CL/*n*BuOH/1,4-*bis*[Bim][PF_6_] from 200/1.0/0.25–4.0 to 700/1.0/0.25–4.0 by mol%. The result found that the optimal ratio of CL/*n*BuOH/1,4-*bis*[Bim][PF_6_] 400/1.0/0.5 mol% at 120 °C for 72 h led to the polymerization conversions higher than 95%, with the molecular weight (*M*_w_) of PCL 20,130 g mol^−1^ (*Đ*~1.80). The polymerization rate of CL increased with the decreasing linker chain length of ionic liquids. Moreover, the mechanistic study was investigated by DFT using B3LYP (6–31G(d,p)) as basis set. The most plausible mechanism included the stepwise and coordination insertion in which the alkoxide insertion step is the rate-determining step.

## 1. Introduction

Biodegradable polymeric materials with optimized physico-chemical and degradation properties have become more important in terms of new biomedical technologies in recent decades. As a result, a new generation of synthetic biodegradable polymers and comparable natural polymers have been modified specifically for biomedical applications [1,2]. Poly(*ε*-caprolactone) (PCL) is one of biologically relevant aliphatic polyesters widely used in various applications. Recently, Woodruff and Hutmacher reported the wide range of biomedical applications involving PCL, including drug delivery, medical devices, and tissue engineering of bone, cartilage, blood vessels, skin, and nerve [3,4,5].

Typically, the synthesis of biodegradable polymers, such as polylactones and polylactide, can be synthesized by several methods, e.g., polycondensation of hydroxycarboxylic acids or ring-opening polymerization (ROP) of cyclic ester monomers. However, the condensation polymerization posts many crucial problems; for example high reaction temperature, accurate stoichiometry, and the removal of by-product of low molecular weight (e.g., water) are expected in ROP [6,7]. Moreover, the polycondensation does not afford high molecular weight polymers [8]. The ROP can be prepared by several methods, such as bulk, solution, and emulsion or dispersion polymerizations [9]. In addition, the initiator or catalyst for ROP is essential to initiate or activate the polymerization of cyclic esters to obtain high molecular weight polyesters with low polydispersity. These general initiators or catalysts for ROP are alkoxides based on metals, such as Al [10,11,12,13,14,15,16,17,18,19,20,21], Sn [11,22,23,24,25,26,27], Zr [11,12,22,23,28], Zn [11,23,29,30,31,32], Sb [11,22], and rare earth or lanthanide metals [11,33,34,35,36].

Generally, the organometallic catalysts used in the synthesis of polyesters *via* condensation polymerization or ring-opening polymerization (ROP) inevitably remain as metal residues in the synthetic polymer products. One of the problems of typical polymerization with an organometallic catalyst is the removal of residual metal from the synthesized polyesters, which is very expensive and complicated. The residue metals and catalyst are of the great concern for biomedical applications since all medical-grade polymers should contain less than 10 ppm heavy metals and 150 ppm residue catalyst according to ASTM F1925-17 (Standard Specification for Semi-Crystalline Poly(lactide) Polymer and Copolymer Resins for Surgical Implants) [37]. Therefore, in order to solve the residue metal in the synthetic polyesters, using organocatalysts is an alternative.

Ionic liquids (ILs) as organocatalysts are one of the most rapidly growing topics of chemistry research on new materials investigation in the last decade. Conventionally, ILs are entirely composed of cations and anions with the melting points lower than 100 °C [38,39,40]. However, ionic liquids are better described as liquid compounds that display ionic-covalent crystalline structures [41]. Attractive characteristics offered by ILs are high thermal stability, negligible vapor pressure, excellent solubility, and broad miscibility with reactants. The uniqueness of this catalyst lays on the acidity and basicity level flexibility as it can be systematically tailored using different types of cations and anions [40,42,43]. ILs have been widely explored in the various inter-disciplinary research areas in the field of organic synthesis, catalysis, biocatalysis, material science, separation process, sensing system, chemical engineering, medicine, green chemistry, and electrochemistry [44,45,46,47,48]. Furthermore, ionic liquid also plays an important role and is widely studied as a catalyst in polymer synthesis. Some significant characteristics of using ionic liquids in ROPs are practically non-toxic, environmentally benign, and applicable under mild preparative conditions. Moreover, better molecular weight control and consistent polymer molecular weight distribution can be afforded by utilizing ILs in ROPs. The ring-opening polymerization (ROP) of 1,4-dioxan-2-one (PDO) catalyzed by ionic liquid 1-butyl-3-methylimidazolium hexafluorophosphate ([Bmim][PF_6_]) coated lipase was investigated. By coating Novozym-435 with 10 wt% ionic liquid [Bmim][PF_6_] (based on PDO) for 6 h, poly(1,4-dioxan-2-one) (PPDO) with a maximum molecular weight (*M*_w_) of 182,100 g mol^−1^ was obtained [49]. 1-Allyl-3-methylimidazolium chloride ([Amim][Cl]) has been used for the dissolution of starch and the homogeneous ring-opening graft polymerization (ROGP) of *ε*-caprolactone (CL) onto starch granules. The ROGP was carried out smoothly and the grafting efficiency of PCL reached 24.42% when the ROGP proceeded at 110 °C for 28 h [50]. Kadokawa and colleagues (2002) examined the ring-opening polymerization of ethylene carbonate using two ionic liquids, 1-butyl-3-methylimidazolium chloroaluminate ([Bmim][Cl]-AlCl_3_) and 1-butyl-3-methylimidazolium chlorostannate ([Bmim][Cl]-SnCl_2_) melts, as polymerization catalysts. The contents of carbonate units and *M*_n_ values of the products obtained using [Bmim][Cl]-SnCl_2_ at 120 °C were higher than those of which derived from [Bmim][Cl]-AlCl_3_ at the same temperature [51]. 1-Butyl-3-methylimidazolium tetrafluoroborate ([Bmim]BF_4_) and 1-butyl-3-methylimidazolium chlorozincate ([Bmim]Cl–(ZnCl_2_)_x_, where *x* is the ZnCl_2_ molar fraction in the mixture, were used as polymerization catalysts in the ring-opening polymerization of ethylene carbonate [52]. In 2015, Kaoukabi and coworkers defined a new method for obtaining poly(*ε*-caprolactone) using 1-butyl-3-methylimidazolium hexafluorophosphate [Bmim][PF_6_] and 1-methyl-3-methylimidazolium hexafluorophosphate [Memim][PF_6_] as a metal-free catalyst with good conversions and narrow molecular weight distributions [53]. According to the aforementioned details, imidazolium ionic liquid has been commonly used as green solvent and catalyst in the ROP of cyclic monomers. These have a high catalytic efficiency in ROP.

In this work, we aim to investigate the synthesis, characterization, and more detail of mechanistic studies of the bulk ROP of *ε*-caprolactone (CL, **1**) with the efficient dicationic ionic liquid catalyst system of 1,n-*bis*[*N*-(*N*′-butylimidazolium)]alkane *bis*hexafluorophosphate (1,n-*bis*[Bim][PF_6_], n = 4, 6, 8 and 10) catalysts (**2a**–**2d**) as a metal-free catalysts and alcohols for the generation of PCL (**3**) of controlled molecular weight and polydispersity (*Đ*) (Figure 1). In addition, we aim to understand the mechanistic aspect of the bulk ROP studies of CL by means of DFT calculations.

## 2. Materials and Methods

### 2.1. Materials

*ε*-Caprolactone (CL, Sigma-Aldrich, St. Louis, MO, USA, 97.0%) was purified by vacuum distillation and the constant boiling fraction from 74–75 °C under reduced pressure were collected. The purified CL was a clear colorless liquid and stored at room temperature under nitrogen atmosphere prior to use in the polymerization process. Imidazole (C_3_H_4_N_2_, Sigma-Aldrich, 99.0%) and chloroform (CHCl_3_, RCI Labscan, Bangkok, Thailand, 99.8%) were used without further purification. 1-Butanol (C_4_H_10_O, Sigma-Aldrich, 98%), 1-bromobutane (C_4_H_9_Br, Sigma-Aldrich, 99.0%), methanol (CH_3_OH, RCI Labscan, 99.9%), dichloromethane (CH_2_Cl_2_, commercial grade), and hexane (C_6_H_14_, commercial grade) were purified by simple distillation prior to use. 1-Dodecanol (C_12_H_26_O, BDH), 1,4-dibromobutane (C_4_H_8_Br_2_, Acros Organic, Geel, Belgium, 99.0%), 1,6-dibromohexane (C_6_H_12_Br_2_, Sigma-Aldrich, 96%), 1,8-dibromooctane (C_8_H_16_Br_2_, Sigma-Aldrich, 98%), 1,10-dibromodecane (C_10_H_20_Br_2_, Sigma-Aldrich, 97%), and stannous actuate (C_16_H_30_O_4_Sn, Sigma-Aldrich, 92.5–100%) were purified by vacuum distillation before use. Acetonitrile (C_2_H_3_N, RCI Labscan, 99.7%) was dried by simple distillation over calcium hydride (CaH_2_). Tetrahydrofuran ((CH_2_)_4_O, Carlo Erba, Cornaredo, Italy, 99.5%) was dried by simple distillation over sodium (Na) metal and benzophenone indicator (C_13_H_10_O, Acros Organic, 99.0%) prior to use.

### 2.2. Instruments

All melting points were determined on a Gallenkamp Electrothermal apparatus, UK. Fourier transform infrared (FTIR) spectra were recorded on a Bruker TENSOR 27 spectrometer (Karlsruhe, Germany) and the wavenumbers were recorded from 400 to 4000 cm^−1^. The high-resolution mass spectra (HRMS, *m*/*z* values) were determined from the HR-TOF-MS Micromass model VQ-TOF2 (Manchester, UK) and Finnigan MAT 95 mass spectrometers (Waltham, MA, USA). The proton and carbon-nuclear magnetic resonance spectroscopy (500 MHz ^1^H-NMR and 125 MHz ^13^C-NMR) were recorded on a Bruker NEO^TM^ 500 NMR spectrophotometer (Karlsruhe, Germany). Tetramethylsilane (TMS), or the residual signals, was used as an internal standard with the solvent resonance as the internal standard (CHCl_3_ impurity in CDCl_3_, *δ* 7.26 and 77.0 ppm; MeOH-*d*_4_, 3.31 and 49.00; DMSO-*d*_6_, 2.50 and 39.52 ppm). The NMR data was reported in the following order: chemical shift, multiplicity and coupling constants (*J*, in hertz). Splitting patterns shown in NMR data were assigned as follows: singlet (*s*), doublet (*d*), triplet (*t*), and multiplet (*m*). The broad NMR peak was denoted by *br* prior to the chemical shift multiplicity. Gel permeation chromatograph (GPC) was carried out on a Waters 2414 GPC (Minneapolis, MN, USA) connected with the refractive index (RI) detector and equipped with Styragel HR5E 7.8 × 300 mm column (molecular weight ranging from 2000–4,000,000 g mol^−1^). THF was used as eluent with a flow rate of 1.0 mL/min at 40 °C. The thermal stability of dicationic liquids was investigated by thermogravimetric analysis (TGA) on a TG-DTA8122 thermo plus EVO2 (Tokyo, Japan). The thermal property of polymer samples was analyzed by the differential scanning calorimetry (DSC) on a METTLER-TOLEDO DSC-1 system (San Francisco, CA, USA).

### 2.3. Synthesis and Characterization of Dicationic Ionic Liquids (DILs, ***2a**–**2d***)

The overview for the synthesis of the DILs was illustrated in Figure 2. The DILs (**2a**–**2d**) were prepared from the reactions of imidazole (**4**) with *bis*-bromoalkanes (**5a**–**5d**). These reactions afforded 1,n-*bis*(1*H*-imidazol-1-yl)alkane (1,n-*bis*[Bim], **6a**–**6d**) that were further reacted with *n*-butyl bromide (*n*BuBr). From this, the 1,n-*bis*[*N*-(*N*′-butylimidazolium)]alkane *bis*bromide salts (1,n-*bis*[Bim][Br], **7a**–**7d**) were formed and reacted with potassium hexafluorophosphate (KPF_6_). From the reaction of 1,n-*bis*[Bim][Br] (**7a**–**7d**) with KPF_6_, the dicationic ionic liquids (DILs, **2a**–**2d**) were obtained. More details in the synthesis of compounds **2**, **6**, and **7** are described in the following.

#### 2.3.1. Synthesis of 1,n-*bis*(1*H*-imidazol-1-yl)alkane (1,n-*bis*[Bim], **6a**–**6d**)

##### Synthesis of 1,4-*bis*(1*H*-imidazol-1-yl)butane (1,4-*bis*[Bim], **6a**)

1,4-*Bis*(imidazole-1-yl)butane (**6a**) was prepared from imidazole, NaOH, and 1,4-dibromobutane as starting materials, with minor modifications as *condition A*, according to the published literature [54,55]. To a 250 mL round bottom flask equipped with a Dean-Stark apparatus, a reflux condenser, and a magnetic stir bar placed on an oil bath equipped with a magnetic stirrer, Imidazole (**4**) (5.00 g, 73.45 mmol) and NaOH (2.93 g, 73.26 mmol) were added with 40 mL of DMSO and stirred at 150 °C for 2 h. Then, the solution of 1,4-dibromobutane (**5a**) (36.73 mmol) in DMSO (10 mL) was slowly added into the previous reaction mixture at 80 °C via a dropping funnel. The reaction mixture was vigorously stirred for 24 h at 100 °C. Then, the reaction was filtered to remove a by-product as solid NaBr and washed with toluene (10 mL). The filtrate was evaporated to dryness and subsequently recrystallized from deionized water to obtain 1,4-*bis*(1*H*-imidazol-1-yl)butane (**6a**) in 5.45 g (78% yield).

1,4-*Bis*[Bim] (**6a**): white needle crystal; mp: 79.2–81.9 °C; *ν*_max_ (thin film): 2937 (C−H), 1653 (C=C), 1609 (C=N), 1231 (C−N) cm^−1^; *δ*_H_ (500 MHz, MeOH-*d*_4_) 1.77 (*m*, 4H, NCH_2_(C*H_2_*)_2_CH_2_N), 4.05 (*m*, 4H, NC*H_2_*(CH_2_)_2_C*H_2_*N), 6.98 (*br s*, 2H, *H*C=CHNCH_2_, imidazole ring), 7.12 (*br s*, 2H, HC=C*H*NCH_2_, imidazole ring), 7.65 (*s*, 2H, N=C*H*−N, imidazole ring); *δ*_C_ (125 MHz, MeOH-*d*_4_) 29.1, 47.3, 120.5, 129.2, 138.4; HRMS (ESI) calcd for C_10_H_15_N_4_ [M+H]+: *m*/*z* 191.1297, found 191.1293.

##### General Procedure for Synthesis of 1,n-*bis*[*N*-(*N*′-butylimidazolium)]alkane (**6b–6d**, n = 6, 8, and 10)

To a 250 mL round bottom flask containing 60% sodium hydride dispersion in mineral oil (3.85 g, 88.14 mmol) in THF (10 mL), the solution of imidazole (**4**) (5.00 g, 73.45 mmol) in THF (20 mL) was slowly added at 0 °C for 1 h. 1,6-Dibromohexane (**5b**) (5.7 mL, 36.73 mmol) was added at room temperature, and the mixture was further refluxed for 72 h, according to published literature [56,57] as *condition B*. After the reaction was complete, the resulting solution was filtered to remove NaBr and washed with THF. The filtrate was evaporated to obtain 1,6-*bis*(1*H*-imidazol-1-yl)hexane (1,6-*bis*[Bim]) (**6b**) in 6.82 g (85% yield).

1,6-*Bis*[Bim] (**6b**): yellow viscous liquid; *ν*_max_ (thin film) 3110–2858 (C−H), 1663 (C=C), 1511 (C=N), 1232 (C−N) cm^−1^; *δ*_H_ (500 MHz, CDCl_3_) 1.23 (*m*, 4H, N(CH_2_)_2_(C*H_2_*)_2_(CH_2_)_2_N), 1.69 (*m*, 4H, NCH_2_C*H_2_*(CH_2_)_2_C*H_2_*CH_2_N), 3.84 (*m*, 4H, NC*H_2_*(CH_2_)_4_C*H_2_*N), 6.82 (*br s*, 2H, *H*C=CHNCH_2_, imidazole ring), 6.98 (*br s*, 2H, HC=C*H*NCH_2_, imidazole ring), 7.37 (*s*, 2H, N=C*H*−N, imidazole ring); *δ*_C_ (125 MHz, MeOH-*d*_4_) 27.0, 31.0, 31.9, 47.8, 120.5, 129.0, 138.4; HRMS (ESI) calcd for C_12_H_19_N_4_ [M+H]+: *m*/*z* 219.1610, found 219.1603.

##### Synthesis of 1,8-*bis*(1*H*-imidazol-1-yl)octane (1,8-*bis*[Bim], **6c**)

1,8-Bis(1H-imidazol-1-yl)octane (1,8-bis[Bim], 6c) was obtained in 7.87 g (87% yield) from the reaction of sodium hydride 60% dispersion in mineral oil (3.85 g, 88.14 mmol), imidazole (4) (5.00 g, 73.45 mmol), and 1,8-dibromooctane (5c) (6.8 mL, 36.73 mmol) following the general procedure.

1,8-*Bis*[Bim] (**6c**): yellow viscous liquid (87% yield); *ν*_max_ (thin film) 2849 (C−H), 1671 (C=C), 1512 (C=N), 1229 (C−N) cm^−1^; *δ*_H_ (500 MHz, CDCl_3_) 1.20 (*m*, 8H, N(CH_2_)_2_(C*H_2_*)_4_(CH_2_)_2_N), 1.67 (*m*, 4H, NCH_2_C*H_2_*(CH_2_)_4_C*H_2_*CH_2_N), 3.84 (*t*, *J* = 7.1 Hz, 4H, NC*H_2_*(CH_2_)_6_C*H_2_*N), 6.83 (*br s*, 2H, *H*C=CHNCH_2_, imidazole ring), 6.97 (*br s*, 2H, HC=C*H*NCH_2_, imidazole ring), 7.38 (*s*, 2H, N=C*H*−N, imidazole ring); *δ*_C_ (125 MHz, MeOH-*d*_4_) 27.4, 30.0, 32.0, 48.0, 120.5, 128.9, 138.3; HRMS (ESI) calcd for C_14_H_23_N_4_ [M+H]+: *m*/*z* 247.1923, found 247.1910.

##### Synthesis of 1,10-*bis*(1*H*-imidazol-1-yl)decane (1,10-*bis*[Bim], **6d**)

1,10-Bis(1H-imidazol-1-yl)decane (1,10-bis[Bim], 6d) was obtained in 8.87 g (88% yield) from the reaction of sodium hydride 60% dispersion in mineral oil (3.85 g, 88.14 mmol), imidazole (4) (5.00 g, 73.45 mmol), and 1,10-dibromoodecane (5d) (8.2 mL, 36.73 mmol) following the general procedure.

1,10-*Bis*[Bim] (**6d**): yellow viscous liquid; *ν*_max_ (thin film) 3109–2852 (C−H), 1670 (C=C), 1511 (C=N), 1230 (C−N) cm^−1^; *δ*_H_ (500 MHz, CDCl_3_) 1.23 (*m*, 12H, N(CH_2_)_2_(C*H_2_*)_6_(CH_2_)_2_N), 1.72 (*m*, 4H, NCH_2_C*H_2_*(CH_2_)_6_C*H_2_*CH_2_N), 3.88 (*t*, *J* = 7.1 Hz, 4H, NC*H_2_*(CH_2_)_8_C*H_2_*N), 6.86 (*br s*, 2H, *H*C=CHNCH_2_, imidazole ring), 7.00 (*br s*, 2H, HC=C*H*NCH_2_, imidazole ring), 7.41 (*s*, 2H, N=C*H*−N, imidazole ring); *δ*_C_ (125 MHz, MeOH-*d*_4_) 27.5, 30.1, 30.4, 32.1, 48.0, 120.5, 129.0, 138.3; HRMS (ESI) calcd for C_16_H_27_N_4_ [M+H]+: *m*/*z* 275.2236, found 275.2230.

#### 2.3.2. Synthesis of 1,n-*bis*[*N*-(*N*′-butylimidazolium)]alkane *bis*bromide salts (1,n-*bis*[Bim] [Br], **7a–7d**)

##### General Procedure for Synthesis of 1,4-*bis*[*N*-(*N*′-butylimidazolium)]butane *bis*bromide salts (1,4-*bis*[Bim][Br], **7a**)

To a 100 mL round-bottomed flask equipped with a reflux condenser, a drying tube, and a magnetic stirrer, 1,4-bis(1H-imidazol-1-yl)butane (6a) (5.00 g, 26.28 mmol) was dissolved in dry acetonitrile (30.0 mL), and 1-bromobutane (6.5 mL, 60.44 mmol) was then added, respectively. The reaction mixture was heated at 98 °C for 72 h. After reaching the reaction time, the reaction was evaporated to yield 1,4-bis[Bim][Br] (7a) in 10.25 g (84% yield).

1,4-*Bis*[Bim][Br] (**7a**): yellow-brown viscous liquid; *ν*_max_ (thin film) 3516–3409 (H−Br, C−H stretching) 3080–2851 (C−H), 1650 (C=C), 1561 (C=N), 1162 (C−N) cm^−1^; *δ*_H_ (500 MHz, MeOH-*d*_4_) 1.00 (*t*, *J* = 7.4 Hz, 6H, C*H_3_*), 1.40 (*m*, 4H, C*H_2_*CH_3_), 1.93 (*m*, 4H, NCH_2_(C*H_2_*)_2_CH_2_N), 2.03 (*m*, 4H, C*H_2_*C_2_H_5_), 4.31 (*t*, *J* = 7.4 Hz, 4H, C*H_2_*C_3_H_7_), 4.39 (*m*, 4H, NC*H_2_*(CH_2_)_2_C*H_2_*N), 7.73 (*br t*, 2H, HC=C*H*NCH_2_, imidazole ring), 7.77 (*br t*, 2H, *H*C=CHNCH_2_, imidazole ring), 9.24 (*s*, 2H, N=C*H*−N, imidazole ring); *δ*_C_ (125 MHz, MeOH-*d*_4_) 13.7, 20.4, 27.7, 32.8, 50.0, 50.7, 123.6, 123.8, 137.0; HRMS (ESI) calcd for C_18_H_32_N_4_Br [M−Br]^+^: *m*/*z* 383.1805, found 383.1790.

##### Synthesis of 1,6-*bis*[*N*-(*N*′-butylimidazolium)]hexane *bis*bromide salts (1,6-*bis*[Bim][Br], **7b**)

Following the general procedure in the aforementioned section, 1,6-bis(imidazole-1-yl)hexane (6b) (5.00 g, 22.90 mmol) was reacted with 1-bromobutane (5.7 mL, 52.67 mmol) in acetonitrile, to afford 1,6-bis[Bim][Br] (7b) in 10.15 g (90% yield).

1,6-*Bis*[Bim][Br] (**7b**): yellow-brown viscous liquid; *ν*_max_ (thin film) 3550–3300 (H−Br), 3080–2851 (C−H), 1650 (C=C), 1561 (C=N) and 1162 (C−N) cm^−1^; *δ*_H_ (500 MHz, MeOH-*d*_4_) 0.99 (*t*, *J* = 7.4 Hz, 6H, C*H_3_*), 1.18–1.60 (*m*, 8H, C*H_2_*CH_3_/N(CH_2_)_2_(C*H_2_*)_2_(CH_2_)_2_N), 1.76–2.04 (*m*, 8H, C*H_2_*C_2_H_5_/NCH_2_C*H_2_*(CH_2_)_2_C*H_2_*CH_2_N), 4.13–4.42 (*m*, 8H, C*H_2_*C_3_H_7_/NC*H_2_*(CH_2_)_4_C*H_2_*N), 7.69 (*br d*, 2H, HC=C*H*NCH_2_, imidazole ring), 7.72 (*br s*, 2H, *H*C=CHNCH_2_, imidazole ring), 9.19 (*s*, 2H, N=C*H*−N, imidazole ring); *δ*_C_ (125 MHz, MeOH-*d*_4_) 13.7, 20.4, 26.3, 33.0, 50.6, 123.7, 136.8; HRMS (ESI) calcd for C_20_H_36_N_4_Br [M−Br]^+^: *m*/*z* 411.2118, found 411.2100.

##### Synthesis of 1,8-*bis*[*N*-(*N*′-butylimidazolium)]octane *bis*bromide salts (1,8-*bis*[Bim][Br], **7c**)

Following the general procedure in the aforementioned section, 1,8-bis(imidazole-1-yl)octane (6c) (5.00 g, 20.30 mmol), 1-bromobutane (5.0 mL, 46.69 mmol) in acetonitrile, to afford 1,8-bis[Bim][Br] (7c) in 9.51 g (90% yield).

1,8-*Bis*[Bim][Br] (**7c**); yellow-brown viscous liquid; *ν*_max_ (thin film) 3550–3200 (H−Br), 3109–2855 (C−H), 1653 (C=C), 1511 (C=N), 1231 (C−N) cm^−1^; *δ*_H_ (500 MHz, MeOH-*d*_4_) 1.00 (*t*, *J* = 7.4 Hz, 6H, C*H_3_*), 1.41 (*m*, 12H, C*H_2_*CH_3_/N(CH_2_)_2_(C*H_2_*)_4_(CH_2_)_2_N), 1.91 (*m*, 8H, C*H_2_*C_2_H_5_/NCH_2_C*H_2_*(CH_2_)_4_C*H_2_*CH_2_N), 4.28 (*t*, *J* = 7.4 Hz, 8H, C*H_2_*C_3_H_7_/NC*H_2_*(CH_2_)_6_C*H_2_*N), 7.69 (*br t*, 2H, HC=C*H*NCH_2_, imidazole ring), 7.72 (*br t*, 2H, *H*C=CHNCH_2_, imidazole ring), 9.17 (*s*, 2H, N=C*H*−N, imidazole ring); *δ*_C_ (125 MHz, MeOH-*d*_4_) 13.7, 20.4, 26.9, 29.5, 30.9, 33.0, 50.6, 50.8, 137.0; HRMS (ESI) calcd for C_22_H_40_N_4_Br [M−Br]^+^: *m*/*z* 439.2431, found 439.2390.

##### Synthesis of 1,10-*bis*[*N*-(*N*′-butylimidazolium)]decane *bis*bromide salts (1,10-*bis*[Bim][Br], **7d**)

Following the general procedure in the aforementioned section, 1,10-bis(imidazole-1-yl)decane (6d) (5.00 g, 18.22 mmol), 1-bromobutane (4.5 mL, 41.91 mmol) in acetonitrile, to obtain 1,10-bis[Bim][Br] (7d) in 9.28 g (93% yield).

1,10-*Bis*[Bim][Br] (**7d**): yellow-brown viscous liquid; *ν*_max_ (thin film): 3600–3300 (H−Br), 3109–2858 (C−H), 1654 (C=C), 1512 (C=N), 1232 (C−N) cm^−1^; *δ*_H_ (500 MHz, MeOH-*d*_4_) 0.99 (*t*, *J* = 7.4 Hz, 6H, C*H_3_*), 1.36 (*m*, 16H, C*H_2_*CH_3_/N(CH_2_)_2_(C*H_2_*)_6_(CH_2_)_2_N), 1.91 (*m*, 8H, C*H_2_*C_2_H_5_/NCH_2_C*H_2_*(CH_2_)_6_C*H_2_*CH_2_N), 4.30 (*m*, 8H, C*H_2_*C_3_H_7_/NC*H_2_*(CH_2_)_8_C*H_2_*N), 7.75 (*br s*, 4H, HC=C*H*NCH_2_/*H*C=CHNCH_2_, imidazole ring); *δ*_C_ (125 MHz, MeOH-*d*_4_) 13.7, 20.4, 27.2, 29.8, 30.1, 31.0, 33.0, 50.6, 50.8, 123.7; HRMS (ESI) calcd for C_24_H_44_N_4_Br [M−Br]^+^: *m*/*z* 467.2744, found 467.2720.

#### 2.3.3. Synthesis of 1,n-*bis*[*N*-(*N*′-butylimidazolium)]alkane *bis*hexafluorophosphates (1,n-*bis*[Bim][PF_6_], **2a–2d**)

##### General Procedure for Synthesis of 1,4-*bis*[*N*-(*N*′-butylimidazo-lium)]butane *bis*hexafluorophosphates (1,4-*bis*[Bim][PF_6_], **2a**)

1,4-*Bis*[*N*-(*N*′-butylimidazolium)]butane *bis*(bromide) (1,4-*bis*[Bim][Br], **6a**) (5.00 g, 10.76 mmol) and KPF_6_ (5.9 g, 32.28 mmol) were dissolved with deionized water, 100 mL in a 250 mL round-bottomed flask equipped with a drying tube and a magnetic stirrer. The reaction mixture was stirred at room temperature for 24 h. Then, the reaction mixture was extracted by CH_2_Cl_2_ for several times to determine bromide ion residue in the organic phase using the AgNO_3_ solution (0.1 M). Finally, the organic layer was dried over Na_2_SO_4_, filtered, and evaporated to obtain 1,4-*bis*[Bim][PF_6_] (**2a**) in 3.53 g (55% yield).

1,4-*Bis*[Bim][PF_6_] (**2a**): white solid; mp: 51.6–53.1 °C; *ν*_max_ (thin film) 3670–3593 (H−F−PF_5_^−^), 3170–2876 (C−H), 1700 (C=C), 1567 (C=N), 1164 (C−N) and 842–559 (P−F) cm^−1^; *δ*_H_ (500 MHz, MeOH-*d*_4_), 0.90 (*t*, *J* = 7.4 Hz, 6H, C*H_3_*), 1.37 (*m*, 4H, C*H_2_*CH_3_), 1.87–1.93 (*m*, 8H, C*H_2_*CH_2_CH_3_/NCH_2_C*H_2_*), 4.19 (*t*, *J* = 7.4 Hz, 4H, NC*H_2_*C_3_H_7_), 4.25 (*br t*, 4H, NC*H_2_*(CH_2_)_2_C*H_2_*N), 7.56 (*br s*, 2H, HC=C*H*NCH_2_, imidazole ring), 7.60 (*br s*, 2H, *H*C=CHNCH_2_, imidazole ring), 8.80 (*s*, 2H, N=C*H*−N, imidazole ring); *δ*_C_ (125 MHz, MeOH-*d*_4_) 13.7, 18.8, 26.1, 31.3, 48.1, 48.7, 122.4, 122.6, 136.0; HRMS (ESI) calcd for C_18_H_32_F_6_N_4_P [M−PF_6_]^+^: *m*/*z* 449.2263, found 449.2290.

##### Synthesis of 1,6-*bis*[*N*-(*N*′-butylimidazolium)]hexane *bis*hexafluorophosphates (1,6-*bis*[Bim][PF_6_], **2b**)

Compound **2b** was synthesized following the general procedure in the aforementioned section: 1,6-*Bis*[*N*-(*N*′-butylimidazolium)]hexane *bis*(bromide) (1,6-*bis*[Bim][Br], **7b**) (5.00 g, 10.16 mmol) and KPF_6_ (5.6 g, 30.48 mmol) were dissolved with in deionized water (100 mL) to afford 1,6-*bis*[Bim][PF_6_] (**2b**) in 3.94 g (62% yield).

1,6-*Bis*[Bim][PF_6_] (**2b**): yellow-brown sticky solid; *ν*_max_ (thin film) 3547–3345 (H−F−PF_5_^–^), 3080–2852 (C−H), 1636 (C=C), 1564 (C=N), 1164 (C−N), 867–519 (P−F) cm^−1^; *δ*_H_ (500 MHz, MeOH-*d*_4_) 0.98 (*t*, *J* = 7.4 Hz, 6H, C*H_3_*), 1.36–1.40 (*m*, 8H, C*H_2_*CH_3_/N(CH_2_)_2_C*H_2_*), 1.88 (*m*, 8H, C*H_2_*C_2_H_5_/NCH_2_C*H_2_*), 4.20 (*m*, 8H, NC*H_2_*C_3_H_7_/NC*H_2_*(CH_2_)_4_C*H_2_*N), 7.71 (*br t*, 2H, HC=C*H*NCH_2_, imidazole ring), 7.75 (*br t*, 2H, *H*C=CHNCH_2_, imidazole ring), 9.22 (*s*, 2H, N=C*H*−N, imidazole ring); *δ*_C_ (125 MHz, MeOH-*d*_4_) 13.7, 20.4, 26.3, 30.5, 32.9, 50.6, 123.6, 123.7, 137.0; HRMS (ESI) calcd for C_20_H_36_F_6_N_4_P [M−PF_6_]^+^: *m*/*z* 477.2576, found 477.2550.

##### Synthesis of 1,8-*bis*[*N*-(*N*′-butylimidazolium)]octane *bis*hexafluorophosphates (1,8-*bis*[Bim][PF_6_], **2c**)

Compound **2c** was synthesized following the general procedure in the aforementioned section: 1,8-*Bis*[*N*-(*N*′-butylimidazolium)]octane *bis*(bromide) (1,8-*bis*[Bim][Br], **7c**) (5.00 g, 9.61 mmol), KPF_6_ (5.3 g, 28.83 mmol) and deionized water (100 mL), to afford 1,8-*bis*[Bim][PF_6_] (**2c**) in 4.25 g (68% yield).

1,8-*Bis*[Bim][PF_6_] (**2c**); yellow-brown sticky solid; *ν*_max_ (thin film) 3650–3200 (H−F−PF_5_^–^), 3081–2857 (C−H), 1652 (C=C), 1563 (C=N), 1165 (C−N), 873–563 (P−F) cm^−1^; *δ*_H_ (500 MHz, MeOH-*d*_4_) 0.98 (*t*, *J* = 7.4 Hz, 6H, C*H_3_*), 1.37 (*m*, 12H, C*H_2_*CH_3_/N(CH_2_)_2_(C*H_2_*)_4_(CH_2_)_2_N), 1.88 (*m*, 8H, C*H_2_*C_2_H_5_/NCH_2_ C*H_2_*(CH_2_)_4_C*H_2_*CH_2_N), 4.20 (*m*, 8H, C*H_2_*C_3_H_7_/NC*H_2_*(CH_2_)_6_C*H_2_*N), 7.61 (*m*, 4H, HC=C*H*NCH_2_/*H*C=CHNCH_2_, imidazole ring), 8.86 (*s*, 2H, N=C*H*−N, imidazole ring); *δ*_C_ (125 MHz, MeOH-*d*_4_) 13.7, 20.4, 26.9, 29.5, 30.8, 33.0, 50.6, 50.8, 123.6, 123.7, 137.0; HRMS (ESI) calcd for C_22_H_40_F_6_N_4_P [M−PF_6_]^+^: *m*/*z* 505.2889, found 505.2880.

##### Synthesis of 1,10-*bis*[*N*-(*N*′-butylimidazolium)]decane *bis*hexafluorophosphates (1,10-*bis*[Bim][PF_6_], **2d**)

Compound **2d** was synthesized following the general procedure in the aforementioned section: 1,10-*Bis*[*N*-(*N*′-butylimidazolium)]decane *bis*(bromide) (1,10-*bis*[Bim][Br], **7d**) (5.00 g, 9.12 mmol), KPF_6_ (5.0 g, 27.36 mmol), and deionized water (100 mL), to afford 1,10-*bis*[Bim][PF_6_] (**2d**) in 4.32 g (70% yield).

1,10-*Bis*[Bim][PF_6_] (**2d**): yellow-brown sticky solid; (70% yield); *ν*_max_ (thin film) 3650–3150 (H−F−PF_5_^–^), 3076–2862 (C−H), 1651 (C=C), 1563 (C=N), 1164 (C−N), 864–641 (P−F) cm^−1^; *δ*_H_ (500 MHz, MeOH-*d*_4_) 0.98 (*t*, *J* = 7.5 Hz, 6H, C*H_3_*), 1.34 (*m*, 16H, C*H_2_*CH_3_/N(CH_2_)_2_(C*H_2_*)_6_(CH_2_)_2_N), 1.88 (*m*, 8H, C*H_2_*C_2_H_5_/NCH_2_C*H_2_*(CH_2_)_6_C*H_2_*CH_2_N), 4.21 (*m*, 8H, C*H_2_*C_3_H_7_/NC*H_2_*(CH_2_)_8_C*H_2_*N), 7.62 (m, 4H, HC=C*H*NCH_2_/*H*C=CHNCH_2_, imidazole ring), 8.87 (*s*, 2H, N=C*H*−N, imidazole ring); *δ*_C_ (125 MHz, MeOH-*d*_4_) 13.7, 20.4, 27.1, 29.8, 30.1, 30.9, 33.0, 50.6, 50.8, 123.7; HRMS (ESI) calcd for C_24_H_44_F_6_N_4_P [M−PF_6_]^+^: *m*/*z* 533.3202, found 533.3180.

### 2.4. Thermal Decomposition Analysis and Cytotoxicity Testing of the Synthesized Dicationic Ionic Liquids (***2a**–**2d***)

The mass loss profile of the synthesized DIL catalysts (**2a**–**2d**) was recorded from 20.0 to 600.0 °C at a heating rate of 10.0 °C/min on a TG-DTA8122 thermo plus EVO2 (Rigaku, Tokyo, Japan), which could obtain the mass loss and mass loss derivative of DILs during thermal decomposition. Heating rates was 10.0 °C/min from 25.0 to 600.0 °C. The synthesized DIL catalysts (**2a**–**2d**) were submitted to preliminary cytotoxicity assays against vero cells (African green monkey kidney fibroblast) at the cytotoxicity test Service Center at Scientific Instruments Center, King Mongkut’s Institute of Technology Ladkrabang University (KMITL), Bankok, Thailand. The synthesized DILs solution at 100 µg/mL concentration were determined using the colorimetric MTT (3-(4,5-dimethylthiazol-2-yl)-2,5-diphenyl tetrazolium bromide) assay in 96-well microtiter plates. DMSO was used as the reference substance. Cell lines were grown at 1 × 10^5^ cell/well and seeded to 96-well plates and further incubated at 37 °C with 5% of CO_2_ atmosphere for 24 h. Then, media was extracted from each well, and 100 µg/mL of sample solution (100 μL/well) was added. After that, 10 µL/well of the MTT solution with concentration of 5 mg/mL was added and incubated at 37 °C with 5% CO_2_ atmosphere for 4 h. Then, the MTT solution was aspirated, and the 100 µL/well of 100% DMSO with 10% of SDS at a ratio 9:1 was added. The quantity of formazan (presumably directly proportional to the number of viable cells) is measured by recording changes in absorbance at 570 nm. The percentages of cytotoxicity were calculated using Equation (1) [58].
(1)$$\%\mathrm{Cytotoxicity}=\left[\frac{A-B}{A}\right]\times 100,$$
where A is the control wells’ absorbance (wells with cells in cultured food), and B is the absorbance of the wells containing the cells. The values of A and B must be subtracted by the absorbance of blank (well with DMSO and SDS solution).

### 2.5. Synthesis of Poly(ε-Caprolactone) via ROP of ε-Caprolactone using the Synthesized DILs (***2a**–**2d***) and Sn(Oct)_2_ as Catalyst with Alcohols Initiator

First of all, purified CL (**1**) (10.00 g, 87.61 mmol) and 1,4-*bis*[Bim][PF_6_] (**2a**) catalyst (0.25–4.00 mol%) were weighed into a 25 mL round bottom flask and followed by the addition of 1-butanol (*n*BuOH) initiator (0.25–1.00 mol%) under N_2_ atmosphere. The reaction flasks were immersed into a preheated oil bath at a temperature of 120 °C for 72 h. The optimized synthetic condition obtained from the ROP of CL with 1,4-*bis*[Bim][PF_6_] (**2a**) and *n*BuOH was applied to other 1,n-*bis*[Bim][PF_6_] catalysts. For those of 1,n-*bis*[Bim][PF_6_] (**2b**–**2d**) and Sn(Oct)_2_ catalysts with the identical polymerization time to 1,4-*bis*[Bim][PF_6_] (**2a**), 1-dodecanol (1.0 mol%) was used as an initiator for the synthesis of PCL at the higher synthetic temperatures of 150, 160, and 170 °C. The obtained crude poly(*ε*-caprolactone) (PCL, **3**) was then purified by precipitation in cold methanol, and the chemical structure and molecular weight averages of the purified PCL was further characterized by ^1^H-NMR and GPC techniques. The thermal property of PCLs was investigated by DSC technique. 5 mg of PCL sample was weighed into the aluminum pan and sealed. The sample was heated from 20 to 250 °C at a heating rate 10 °C/min and then held at 250 °C for 1 min. Then, the molten sample was cooled down from 250 to 20 °C at a cooling rate of 10 °C/min. For the second heating, the sample was heated from 20 to 250 °C at a heating rate 10 °C/min.

### 2.6. Kinetic Studies of the ROP of ε-Caprolactone Catalyzed by DILs (***2a**–**2d***) Catalysts with 1-Dodecanol Initiator

Purified CL (**1**) (10.00 g, 87.61 mmol), 1,n-*bis*[Bim][PF_6_] (**2a**–**2d**) catalysts (0.5 mol%) and 1-dodecanol (*n*C_12_H_25_OH) (1.0 mol%) were weighed into a 25 mL round bottom flask under N_2_ atmosphere. The reaction flasks were immersed into a preheated oil bath at a constant temperature of 150 °C. The obtained crude PCL was analyzed by ^1^H-NMR technique using CDCl_3_ as solvent. The polymerization kinetics was investigated by the conventional method of ^1^H-NMR. The monomer conversion for the ROP of CL with 1,n-*bis*[Bim][PF_6_] (**2a**–**2d**) catalysts with *n*C_12_H_25_OH initiator was determined from Equation (2) [53].
(2)$$\%\mathrm{Conversion}=\left[\frac{I_{\alpha }}{I_{\alpha }+I_{\beta }}\right]\times 100,$$
where *I_α_* and *I_β_* are the proton integral from the ethylenic protons of PCL (**3**) (4.14 ppm) and monomer (4.26 ppm), respectively.

### 2.7. Computational Study by Density Functional Theory (DFT)

Density functional theory (DFT) calculations were performed at B3LYP level with 6–31G(d,p) basis set to study the ROP of *ε*-caprolactone (**CL**) triggered by 1,4-*bis*[Bim][PF_6_] (**2a**) catalyst [59,60]. The local lowest point was found via structural optimizations of reactants (**R**), complexes (**COM**), intermediates (**INT**), and products (**P**). Using the usual Berny transition-state optimization method, all transition states (**TS**) were estimated to be saddle points. Furthermore, vibrational frequency calculations were performed at the same level of theory to verify optimal structures by zero imaginary frequency stretching along the reaction path for **R**, **INT**, and **P**, and one imaginary frequency stretching along the reaction path for all **TS** structures. By including zero-point vibration energy contributions, Gibbs free energy profiles for all systems were adjusted. Thermal adjustments were also tested at 298 °F. The Gaussian 09 suite of programs was used for all calculations [61].

## 3. Results and Discussion

### 3.1. Synthesis of Dicationic Ionic Liquids (***2a**–**2d***)

Dicationic ionic liquids, 1,n-*bis*[Bim][PF_6_] (**2a**–**2d**, n = 4, 6, 8, and 10) (Figure 2) were successfully synthesized by nucleophilic substitution as follows: firstly, the synthesis of *bis*(1*H*-imidazol-1-yl)alkane (**6a**–**6d**) were prepared from imidazole (**4**) with dibromoalkane (**5a**–**5d**) via nucleophilic substitution. The product **6a** was recrystallized from deionized water and obtained in 78% yield as a white crystal solid. For compound **6b** (85% yield), **6c** (87% yield) and **6d** (88% yield) were afforded as yellow viscous liquid. The structure **6a**–**6d** were characterized by ^1^H-NMR in MeOH-*d*_4_, which showed an important signal of two *N*-methylene protons (NC*H_2_*) at 4.19–4.31 (*t*, *J* = 7.5 Hz, 4H), and the two methine protons (N–C*H*=N) of imidazoles moiety at 8.80–9.22 (*s*, 2H) ppm (see Appendix A). Then, *bis*[*N*-(*N*′-butylimidazolium)]butane *bis*bromide (**7a**–**7d**) were prepared from *bis*(1*H*-imidazol-1-yl)alkane (**6a**–**6d**) which was reacted with 1-bromobutane by *N*-alkylation reaction. All products are yellow-brown viscous liquids and obtain in high yields (84–93%). The ^1^H-NMR spectrum (500 MHz, in MeOH-*d*_4_) of 1,4-*bis*[Bim][Br] (**7a**) exhibited the characteristic signals of two *N*-methylene protons from *n*-butyl group at 4.31 (*t*, *J* = 7.4 Hz, 4H) and 4.39 (*m*, 4H) of the *N*-methylene proton of alkyl linker (NC*H_2_*CH_2_CH_2_C*H_2_*N). In addition, the methyl protons of *n*-butyl groups showed the chemical shift value at 1.00 (*t*, *J* = 7.4 Hz, 6H) ppm, and the two methine protons (N–C*H*=N) of imidazolium rings exhibited the chemical shift at 9.24 (*br s*, 2H) ppm (see Appendix A). Moreover, the FT-IR spectrum exhibited the important vibration frequency of hydrogen bonding between H atom (N–C*H*=N) and Br counter anion at 3516–3409 cm^−1^ (see Appendix A). DILs, 1,n-*bis*[Bim][Br] (**7b**–**7d**, n = 6, 8 and 10) exhibited both ^1^H-NMR and FT-IR spectra data in similar to those of 1,4-*bis*[Bim][Br] (**7a**) (see Appendix A). Finally, compounds **7a**–**7d** were treated with KPF_6_ in the aqueous solution through bromide-hexafluorophosphate exchange to give *bis*[*N*-(*N*′-butylimidazolium)]alkane *bis*(hexafluorophosphate) (**2a**–**2d**) in 55–70% yield. The structure of **2a**–**2d** was characterized by FT-IR, which showed significant vibration wavenumbers of P−F stretching at 827 cm^−1^, P−F bending at 557 cm^−1^, and H bonding (H−F−PF_5_^−^) at 3721–3610 cm^−1^ (see Appendix A). In addition, ^1^H-NMR (500 MHz, in MeOH-*d*_4_) spectra of **2b**–**2d** (Figure 1) exhibited the characteristic signals of the methine protons (N–C*H*=N) of imidazolium rings at the chemical shifts of 8.80, 9.22 and 8.86 ppm, respectively, which shown on the correspondent spectra as broad singlets. However, the methine proton signal of **2a** is not detectable due to this proton being strongly acidic, resulting in the exchange with labile deuterated proton of MeOH-*d*_4_.

### 3.2. Thermal Decomposition Analysis and Cytotoxicity Testing of the Synthesized DIL Catalysts (***2a**–**2d***)

1,n-*Bis*[Bim][PF_6_] catalysts (**2a**–**2d**) were subjected to the thermal stability investigation by the thermogravimetric analysis (TGA). The obtained TGA curves for the synthesized 1,n-*bis*[Bim][PF_6_] catalysts at a heating rate of 10.0 °C/min are illustrated in Figure 2. From Figure 2, it was found that the onset temperature (*T*_0_) of thermal decomposition of compounds **2b**–**2d** was found at around 260.0 °C. Furthermore, compound **2a** showed the highest thermal stability with *T*_0_ around 320.0 °C. As a results, it clearly demonstrated that the synthesized 1,n-*bis*[Bim][PF_6_] catalysts exhibited relatively high thermal stability. Therefore, it was possible to use these catalysts in the high temperature polymerization of cyclic esters.

The preliminary cytotoxicity evaluation of these DILs (**2a**–**2d**) was tested using the conventional MTT assay. In order to find out whether these DILs (**2a**–**2d**) are toxic to normal cells, their anti-proliferative activity against African green monkey kidney fibroblast (Vero) were evaluated, and the results are shown in Appendix A. Percentage cell viability (% living cells) showed that all DILs (**2a**–**2d**) exhibited less cytotoxicity against tested cell lines with a concentration of 100 µg/mL. Furthermore, the percentage of cell viability from cytotoxicity testing of the synthesized DILs was higher than that of the control experiment of 1% DMSO.

**Figure 1 polymers-13-04290-f001:**
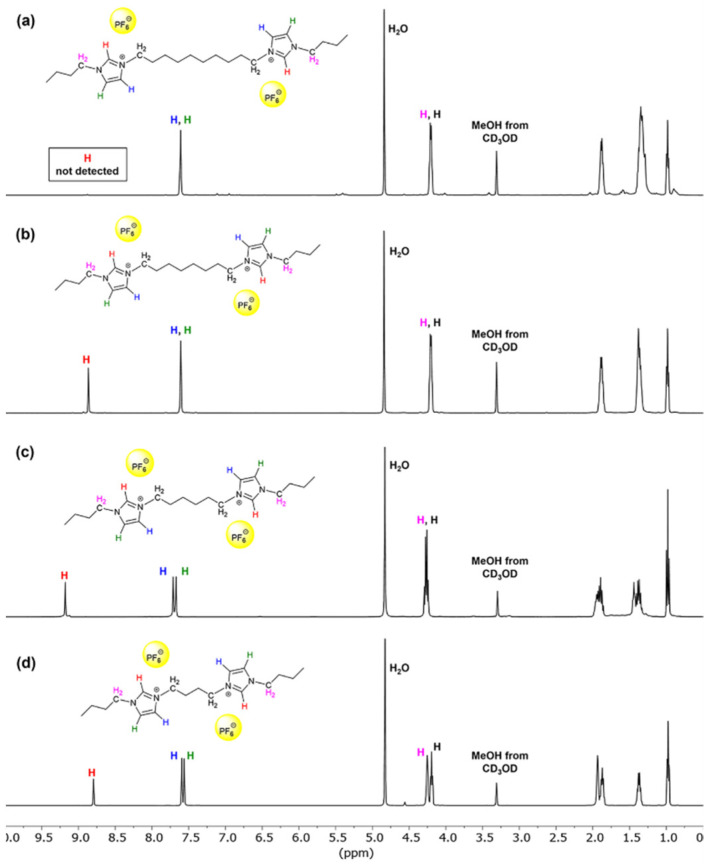
The ^1^H-NMR spectra of the synthesized dicationic ionic liquid compounds in MeOH-*d*_4_: (**a**) 1,10-*bis*[Bim][PF_6_] **2d**, (**b**) 1,8-*bis*[Bim][PF_6_] **2c**, (**c**) 1,6-*bis*[Bim][PF_6_] **2b**, and (**d**) 1,4-*bis*[Bim][PF_6_] **2a**.

**Figure 2 polymers-13-04290-f002:**
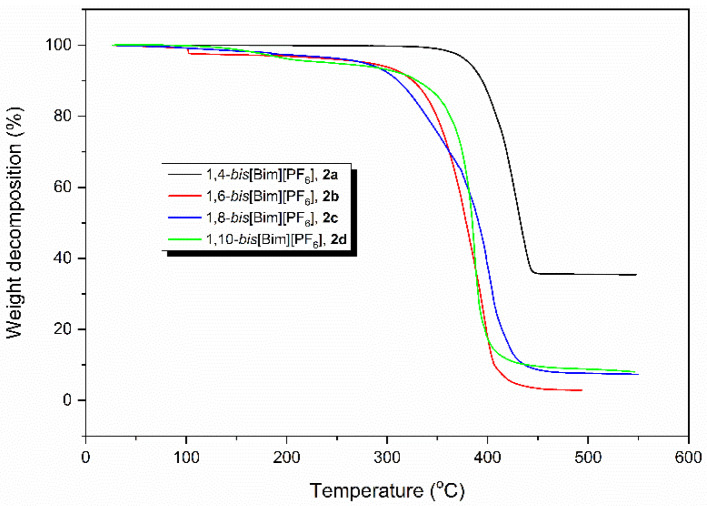
The TGA heating curves for the synthesized 1,n-*bis*[Bim][PF_6_] (**2a**–**2d**) catalysts at a heating rates of 10 °C/min.

### 3.3. Synthesis of Poly(ε-Caprolactone) via the ROP of ε-Caprolactone using the Synthesized 1,4-bis[Bim][PF_6_] (***2a***) Catalyst with 1-Butanol Initiator

Generally, it is known that the hydrogen atom at the C-2 position of the imidazolium salt (H-2) is a weak Brønsted acid that can catalyze many organic reactions. We first evaluated the catalytic activity of 1,4-*bis*[Bim][PF_6_] catalyst (**2a**) in the presence of 1-butanol (*n*BuOH) as an initiator for the ROP of CL (**1**) at 120 °C for 72 h. Furthermore, the catalytic reactivity of compound **2a** in the ROP of CL was also compared to 1,4-*bis*[Bim][Br] (**7a**). The results suggested that the ROP of CL with compound **7a** cannot occur under the synthetic condition used in this work. In 1,4-*bis*[Bim][Br] (**7a**) system, the strong H-bond between Br ion and H atom on the imidazole ring reduced its catalytic efficiency in the ROP of CL [56]. In the case of 1,4-*bis*[Bim][PF_6_] catalyst (**2a**), the polymerization of CL was completed after 48 h at the molar ratios of CL/*n*BuOH/DILs of 200–700/1.0/0.25–4.0. The GPC analysis of molecular weight averages, molecular weight distribution (*Ð*) of the synthesized poly(*ε*-caprolactone) (PCL, **3**) is summarized in Table 1.

From Table 1, it was found that the increasing molar ratio of monomer to catalyst resulted in the increasing of PCL molecular weight. The % conversion for the ROP of CL catalyzed by 1,4-*bis*[Bim][PF_6_] (**2a**) with *n*BuOH initiator was higher than 97% for all CL/*n*BuOH/DILs ratio. Moreover, the molecular weight of PCL seemed to decease with increasing 1,4-*bis*[Bim][PF_6_] (**2a**) concentration. The highest average molecular weight (*M*_w_ = 20,130 g mol^−1^) of PCL was obtained at CL/*n*BuOH/1,4-*bis*[Bim][PF_6_] ratio of 400/1.0/0.50. mol% The molecular weight of PCL showed the increasing trend in the molar ratio of CL/*n*BuOH ranging from 200/1.0 to 400/1.0. However, when the molar ratio of CL/*n*BuOH higher than 400/1.0, the molecular weight of the obtained PCL tended to decrease. Furthermore, the *Ð* values for all synthesized PCLs were in the range of 1.16–1.88, indicating the low amount of transesterification occurred in our polymerization systems. From these obtained results, the most suitable synthetic condition from Table 1 would be applied to other catalysts, as shown in the following section.

### 3.4. Synthesis of Poly(ε-Caprolactone) via the ROP of ε-Caprolactone using the Synthesized DILs (***2a**–**2d***) and Sn(Oct)_2_ Catalyst with 1-Dodecanol Initiator

After obtaining the optimized synthetic condition for PCL (**3**) (at the CL (**1**)/*n*BuOH/1,4-*bis*[Bim][PF_6_] ratio of 400/1.0/0.50 mol%), as in previous section, it was applied to other DILs catalysts and Sn(Oct)_2_. In this section, the 1-dodecanol (*n*C_12_H_25_OH) initiator was used as a substituent for *n*BuOH in the synthesis at a higher temperature range. The ROP of CL catalyzed by DILs (**2a**–**2d**) and Sn(Oct)_2_ with *n*C_12_H_25_OH initiator was carried out at 150, 160, and 170 °C for 72 h. After complete polymerization, the obtained crude PCL was dissolved in CHCl_3_ and precipitated in cold methanol yielding purified PCL. The obtained purified PCL was further characterized by ^1^H-NMR and GPC technique, and the results are summarized in Table 2.

From Table 2, the % conversion for the ROP of CL with all catalysts were higher than 96%. Moreover, it was found that the molecular weight of PCL seemed to be higher than the molecular weight of PCL shown in Table 1. This suggested that the molecular weight of PCL could be improved by increasing the polymerization temperature. Furthermore, the molecular weight of PCL slightly increased with the increasing of DIL chain length. When comparing the performance of DILs with the conventional system of Sn(Oct)_2_, it was found that the DILs showed the equivalent performance to Sn(Oct)_2_ in terms of PCL molecular weight. Additionally, the *Ð* values for the synthesized PCL were higher than PCL shown in Table 1. This clearly demonstrated that the increasing of synthesis temperature could result in the broadening of PCL molecular weight distribution due to the higher amount of transesterification reactions.

The thermal property of the synthesized PCL was subsequently identified by DSC technique. An example of thermal characterization of PCL sample (entry 25) is displayed in supporting data (Appendix A). It was found that the crystalline melting temperature (*T_m_*) of PCL was found at around 40–58 °C for the first heating. From the cooling step, the crystallization exotherm was observed at around 30–40 °C. This indicated that the synthesized PCL could crystallize under the synthetic condition used in this work. For the second heating, the *T_m_* of PCL was found ca. 45–75 °C. It was important that the one melting peak of PCL was observed, suggesting one shape and size of PCL crystal which was obtained.

### 3.5. Kinetic Studies of the ROP of ε-Caprolactone Catalyzed by the Synthesized DIL Catalysts (***2a**–**2d***) with 1-Dodecanol Initiator

The reactivity of all synthesized DILs (**2a**–**2d**) in the ROP of CL was investigated at the temperature of 150 °C for 4 h using the ^1^H-NMR technique. An example of the ^1^H-NMR spectra of crude PCL obtained from the ROP of CL catalyzed by 0.50 mol% of 1,4-*bis*[Bim][PF_6_] (**2a**) and 1.0 mol% of *n*C_12_H_25_OH at 150 °C are illustrated in Figure 3. The ^1^H-NMR spectra of crude PCL were obtained from the ROP of CL catalyzed other DILs (**2b**–**2d**) are depicted in the supporting data (see Appendix A).

From Figure 3, the intensity of triplet proton of the methylene group connected to carbonyl carbon of PCL chain at 2.30 ppm (1 and 6) increased with increasing polymerization time, but the intensity of triplet proton of the methylene group adjacent to carbonyl carbon of CL ring at 2.65 ppm (1′) decreased similar to other DIL catalysts.

Furthermore, all spectra showed the multiplet proton of methyl chain end of butyl group (CH_3_) at 0.90 ppm (a). The multiplet proton of methylene group of PCL and CL was found at 1.20–1.65 ppm (2, 3, 4, 7, 8, 9, 2′, 3′, 4′, b, c). The triplet proton of methylene group connected to –OH end group was found at 3.65 ppm (10). Finally, the triplet proton of methylene groups adjacent to oxygen atom of CL and PCL were found at 4.12 (5, d) and 4.25 ppm (5′), respectively. From the spectra, the monomer conversion was determined from the Equation (2). The plots of % monomer conversion against polymerization time for the ROP of CL catalyzed by 0.50 mol% of DILs (**2a**–**2d**) with 1.0 mol% of *n*C_12_H_25_OH at 150 °C are depicted in Figure 4.

From Figure 4, the % monomer conversion for the ROP of CL catalyzed by 0.50 mol% of 1,4-*bis*[Bim][PF_6_] (**2a**) with 1.0 mol% of *n*C_12_H_25_OH approaches 100% at the lower time than 1,6-*bis*[Bim][PF_6_] (**2b**), 1,8-*bis*[Bim][PF_6_] (**2c**), and 1,10-*bis*[Bim][PF_6_] (**2d**), respectively.

### 3.6. Mechanistic Study Using DFT Method

From the obtained results, the synthesized 1,4-*bis*[Bim][PF_6_] (**2a**) acted as the most efficient catalyst in the ROP of CL in terms of PCL molecular weight and polymerization rate. To investigate the polymerization mechanism, the computation detail by DFT method was utilized. The mechanism of the ROP of CL catalyzed by 1,4-*bis*[Bim][PF_6_] (**2a**) and initiated by *n*BuOH was described and used as the template for other DIL catalysts. In this part, the abbreviations of **COM**, **TS**, **INT**, and **P** were the complex, the transition state, the intermediate, and the product, respectively. The polymerization started with the coordination between CL monomer, **2a**, and *n*BuOH that could be classified into three mechanisms. Starting with the coordination of two units of CL with the acidic protons on the imidazole ring of **2a**, resulting in the formation of **COM1,** is depicted in Figure 3. The coordination of monomer was a spontaneous process that could be confirmed by the determined Gibbs free energy (ΔG) of −45.21 kcal mol^−1^.

Then, **COM1** was further coordinated with *n*BuOH via two steps before the ring-opening of CL ring: (i) the coordination of *n*BuOH with the carbonyl group of CL ring forming **COM2** with G value of -31.57 kcal mol^−1^ and (ii) the coordination of *n*BuOH with the acyl oxygen atom of CL ring forming **COM3** with ΔG value of −31.28 kcal mol^−1^. For the third mechanism, the flexible C−C bond of **2a** could also be rotated to produce **Twisted**-**2a**. The imidazole rings were placed on the same side due to the C−C bond rotation that affected the coordination of monomer. From this, one unit of CL could be used to coordinate **Twisted**-**2a** yielding **COM4** with ΔG of −22.94 kcal mol^−1^. Although the ΔG value of **COM4** (22.94 kcal mol^−1^) is a less negative value than that of **COM2** and **COM3**, it still indicated that the coordination process was spontaneous. Because of the steric hindrance around the imidazole rings, *n*BuOH could only coordinate at the acyl oxygen atom of CL yielding **COM5** with ΔG of −13.74 kcal mol^−1^.

Firstly, we discussed the ROP mechanism in the first pathway that was calculated as the stepwise mechanism as shown in Figure 4. After the coordination of CL with **COM1**, the carbonyl oxygen of CL was coordinated with *n*BuOH, producing the stimulated monomer as **COM2** with the more electrophilic carbonyl carbon. The oxygen atom attacked the weak carbonyl carbon, and the proton of *n*BuOH was transferred to the oxygen atom on the carbonyl carbon via the four-membered ring to form transition state 1 (**TS1**-**8**) with ΔG of 14.64 kcal mol^−1^, indicating the non-spontaneous process and required ΔG^‡^ of 46.24 kcal mol^−1^. After that, the planar carbonyl carbon was rearranged to the tetrahedral geometry yielding intermediate 1 (**INT1**-**9**). At **TS1**-**10**, the proton was transferred to acyl oxygen, and the C−O of acyl bond was broken, and that required the ΔG^‡^ of 25.94 kcal mol^−1^. Finally, the product of C−O bond cleavage was given as product 1 (**P1**-**11**) with the ΔG of −21.82 kcal mol^−1^. The initiation of another CL molecule was also proceeded by a similar mechanism as described. The results of the second initiation revealed that the ΔG of **TS1**-**12** was 18.02 kcal mol^−1^ and need ΔG^‡^ of 39.84 kcal mol^−1^. The ΔG and ΔG^‡^ values of **TS1**-**14** were 17.48 and 29.08 kcal mol^−1^, respectively. From the results of ΔG and ΔG^‡^ for the first and second initiations, it could be concluded that **TS1**-**8** displayed the highest ΔG^‡^. Therefore, the rate-determining step for the ROP of CL catalyzed by the 1,4-*bis*[Bim][PF_6_] (**2a**) with *n*BuOH initiator was the formation of **TS1**-**8**.

The second mechanism for the ROP of CL catalyzed by the 1,4-*bis*[Bim][PF_6_] (**2a**) with *n*BuOH initiator is illustrated in Figure 5. The acyl oxygen atom of CL was coordinated with *n*BuOH after the coordination between CL and **2a** yielding **COM3**. The ΔG of **TS2**-**16** was 16.14 kcal mol^−1^ and required the ΔG^‡^ of 47.42 kcal mol^−1^. Then, the carbonyl carbon of CL was attacked by the oxygen atom of *n*BuOH, and the proton of *n*BuOH was transferred to the acyl oxygen atom of CL, leading to formation of **P2**-**17**. Although the ΔG (18.02 kcal mol^−1^) of the **TS2**-**18** was higher than **TS2**-**16**, **TS2**-**18** showed a lower value of ΔG^‡^ than that of **TS2**-**16**. This indicated that **TS2**-**16** was the rate-determining step for the second ROP mechanism of CL catalyzed by the 1,4-*bis*[Bim][PF_6_] (**2a**) with *n*BuOH initiator.

For the third mechanism, the ROP of CL catalyzed by the 1,4-*bis*[Bim][PF_6_] (**2a**) with *n*BuOH initiator proceeded via the one **TS3**-**20**, as displayed in Figure 6. In this mechanism, the CL monomer was coordinated with **Twisted-2a**, yielding **COM4**. Then, the acyl oxygen atom of the activated CL was coordinated with *n*BuOH, resulting in the **COM5**. Then, the carbonyl carbon of activated CL was attacked by the oxygen atom of *n*BuOH and followed by the acyl bond scission through the four-membered ring to form **TS7** with ΔG and ΔG^‡^ of 29.29 and 44.62 kcal mol^−1^, respectively.

The first and second ROP mechanisms demonstrated that the ΔG values of **COM2** and **COM3** were found to be −31.57 and −31.28 kcal mol^−1^, respectively. However, the ΔG^‡^ values for the rate-determining step of the first and second ROP mechanisms were 46.24 and 47.42 kcal mol^−1^, respectively. For the third ROP mechanism, the **COM5** displayed a higher ΔG value than **COM2** and **COM3**. Furthermore, the ΔG^‡^ value for the third ROP mechanism was the highest. Therefore, based on ΔG and ΔG^‡^ for these three ROP mechanisms, the first ROP mechanism showed the lowest in both of ΔG and ΔG^‡^ values. Therefore, the ROP mechanism of CL catalyzed by the synthesized 1,4-*bis*[Bim][PF_6_] (**2a**) with *n*BuOH initiator occurred and proceeded through the first ROP mechanism. The overall ΔG profile for the three mechanisms of the ROP of CL catalyzed by the synthesized 1,4-*bis*[Bim][PF_6_] (**2a**) with *n*BuOH initiator using the calculation from the B3LYP/6–31G(d,p) level is illustrated in Figure 5.

## 4. Conclusions

The DILs with different chain lengths (**2a**–**2d**) were successfully synthesized using the commercially available imidazole as a starting material with % yield in a range of 55–70%. The synthesized DILs were completely characterized by FITR, ^1^H-NMR, ^13^C-NMR HRMS, and mass spectrometry. From TGA analysis, the synthesized DILs showed high thermal stability with starting degradation temperature of around 260–320 °C. Based on cytotoxicity testing, the synthesized DILs exhibited less cytotoxicity against monkey kidney epithelial cells with a concentration of 100 µg/mL. From bulk polymerization of CL catalyzed by 1,4-*bis*[Bim][PF_6_] (**2a**) with *n*BuOH initiator, the PCL with *M*_w_ of 20,130 g mol^−1^ was obtained at the molar ratio of CL/*n*BuOH/1,4-*bis*[Bim][PF_6_] of 400/1.0/0.50. The chain length of the synthesized DILs slightly affected the molecular weight of PCL. The molecular weight of PCL could be improved by increasing the polymerization temperature. The highest *M*_w_ (32,227 g mol^−1^) of PCL was obtained from the ROP of CL catalyzed by 1,6-*bis*[Bim][PF_6_] (**2b**) with *n*C_12_H_25_OH initiator at 170 °C for 72 h. A kinetics study of the performance of DILs in the ROP of CL by ^1^H-NMR technique was successfully compared at 150 °C. The reactivity of 1,4-*bis*[Bim][PF_6_] (**2a**) in the ROP of CL with 1.0 mol of *n*C_12_H_25_OH was higher than 1,6-*bis*[Bim][PF_6_] (**2b**), 1,8-*bis*[Bim][PF_6_] (**2c**), and 1,10-*bis*[Bim][PF_6_] (**2d**), respectively. The effectiveness of the synthesized DILs was equivalent to the conventional system of Sn(Oct)_2_ under the condition used in this work. From a computational study by DFT using the B3LYP/6–31G(d,p) level, the ROP of CL started by coordination of CL monomer with 1,4-*bis*[Bim][PF_6_] (**2a**), resulting in more electrophilic carbonyl carbon of CL and followed by the attack of *n*BuOH to the carbonyl carbon of CL. Then, the proton of *n*BuOH is transferred to acyl oxygen, resulting in the acyl bond of CL scission. The ROP of CL with other DILs has been proposed through a similar mechanism to 1,4-*bis*[Bim][PF_6_] (**2a**). Finally, the mechanistic results obtained from this work may be applied to describe the catalytic behavior of other organocatalytic systems in the ROP of cyclic esters.

## Data Availability

Not applicable.

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
