# Peer review of "Organocatalytic Ring-Opening Polymerization of ε-Caprolactone Using bis(N-(N′-butylimidazolium)alkane Dicationic Ionic Liquids as the Metal-Free Catalysts: Polymer Synthesis, Kinetics and DFT Mechanistic Study"

_polymers, 2021, doi:10.3390/polym13244290_

Round 1
Reviewer 1 Report
Dear Authors,
The research works is interesting and accept with minor corrections.
Finally, cut out repetitions, run a spell-checker, and have it revised. The article needs a complete alignment.
p1 : Abstract is lengthy and could be modified and reduced.
Abbreviations of all the content inside the figures should be provided, spacing between the figure and figure legends need to be consistent
Author Response
Dear Editor-in-Chief
Enclosed please find the revised manuscript entitled “Organocatalytic ring-opening polymerization of ε-caprolactone using bis(N-(N'-butylimidazolium)alkane dicationic ionic liquids as the metal-free catalysts: Polymer synthesis, kinetics and DFT mechanistic study” (Manuscript ID: polymers-1468962) resubmitted for publication in Polymers.
With reference to the comments and suggestions of the reviewers I have amended the manuscript as follows:
Answer of the reviewer’s comments (polymers-1468962)
Reviewer: 1
- Comments 1:
P1: Abstract is lengthy and could be modified and reduced.
Our response:
We agree with Reviewer#1. We have already shortened the abstract part as commented.
- Comments 2:
Abbreviation of all the content inside the figures should be provided, spacing between the figure and figure legends need to be consistent.
Our response:
We agree with Reviewer#1. We have already added the meaning of the abbreviations shown in all figures as commented. Moreover, the space between figure and figure legend has been carefully checked.
I thank you the referees for very useful comments and I hope the revised manuscript will be favourably considered for publication.
Sincerely yours,
Puttinan Meepowpan, Ph.D.

Reviewer 2 Report
The manuscript (polymers-1468962) presents use of ionic liquids based on 1,n-bis[N-(N'-butylimidazolium)]alkane bishexafluorophosphate as catalysts for the ring opening polymerization of ε-caprolactone. The authors used butanol as initiator. A kinetics and mechanistic investigation were performed. The manuscript is interesting and logically arranged, however before publication I have a couple of remarks/suggestions:
- The manuscript requires extensive English editing in almost every part.
- Scheme 2 – there is no difference between condition A and condition B, please check experimental description 2.3.1.1!!!
- The polydispersity index values are higher than 1.5 which is not specific for ROP, how do the authors account this aspect?
- The legend of Figure 4 is not readable.
- Why in Table 1 for sample 1 and 2 the theoretical molecular weights are different? Same question for sample 2 and 3 in Table 2, considering that same ratio between monomer and initiator is considered?
Author Response
Dear Editor-in-Chief
Enclosed please find the revised manuscript entitled “Organocatalytic ring-opening polymerization of ε-caprolactone using bis(N-(N'-butylimidazolium)alkane dicationic ionic liquids as the metal-free catalysts: Polymer synthesis, kinetics and DFT mechanistic study” (Manuscript ID: polymers-1468962) resubmitted for publication in Polymers.
With reference to the comments and suggestions of the reviewers I have amended the manuscript as follows:
Answer of the reviewer’s comments (polymers-1468962)
Reviewer: 2
- Comments 1:
The manuscript requires extensive English editing in almost every part.
Our response:
We agree with Reviewer#2. We have carefully check the grammar of the manuscript as commented.
- Comments 2:
Scheme 2 there is on different between condition A and condition B, please check experimental description 2.3.1.1.
Our response:
We agree with Reviewer#2. We have carefully check the condition A and condition B shown in Scheme 2 as commented.
- Comments 3:
The polydispersity index value are higher than 1.5 which is not specific for ROP, how do the authors account this aspect?
Our response:
We agree with Reviewer#2. The polydispersity index value of the polymers synthesized from the ROP process can be higher than 1.5. It depends on the type on initiator used and the synthesis conditions. This higher polydispersity index value may be related to the difference in the initiation and propagation rate or the occurrence of the transesterification reaction. In some case, the highly reactive initiator such as tin(II) octoate (Sn(Oct)2) in the solvent-free ROP of CL can also act as the strong transesterification catalyst that cause the high value of polydispersity index [R1]. Furthermore, the ROP of CL with titanium(IV) alkoxide (Ti(OR)4) initiators at 120 °C for 72 h can also yield the PCLs with very high polydispersity index value of 2-3 [R2].
References:
R1. Bero, M., Czapla, B., Dobrzynski, P., Janeczek, H., Kasperczyk, J. Copolymerization of glycolide and ε-caprolactone. Macromol. Chem. Phys. 1999, 200, 911-916.
R2. Meelua, W., Molloy, R., Meepowpan, P., Punyodom, W. Isoconversional kinetic analysis of ring-opening polymerization of caproalctone: Steric influence of titanium(IV) alkoxaides as initaitors. J. Polym. Res. 2012, 19, 9799.
- Comments 4:
The legend of Figure 4 is not readable.
Our response:
We agree with Reviewer#2. We have already edited the legend of Figure 4 as commented.
- Comments 5:
Why in Table 1 for sample 1 and 2 the theoretical molecular weights are different? Same question for sample 2 and 3 in Table 2, considering that same ratio between monomer and initiator is considered?
Our response:
We agree with Reviewer#2. We have recalculated all of the theoretical molecular weights of PCLs shown in Tables 1 and 2. The theoretical molecular weights of PCLs are slightly difference due to their different values of %conversion.
I thank you the referees for very useful comments and I hope the revised manuscript will be favourably considered for publication.
Sincerely yours,
Puttinan Meepowpan, Ph.D.
